

# In vitro anti-diabetic effects and phytochemical profiling of novel varieties of *Cinnamomum zeylanicum* (L.) extracts

W.A. Niroshani M. Wariyapperuma[1], Sagarika Kannangara[2], Yasanandana S. Wijayasinghe[3], Sri Subramanium[4] and Bimali Jayawardena[1]

[1] Department of Chemistry, University of Kelaniya, Kelaniya, Western, Sri Lanka
[2] Department of Plant and Molecular Biology, University of Kelaniya, Kelaniya, Western, Sri Lanka
[3] Department of Biochemistry, University of Kelaniya, Ragama, Western, Sri Lanka
[4] Department of Chemistry, University of North Texas, Texas, United States of America

Corresponding author
Bimali Jayawardena,
bimalimadu123@gmail.com

## ABSTRACT

**Background**. Diabetes mellitus type 2 (DMT2) is a leading metabolic disorder in the world. Anti-diabetic actions of phytochemicals from various medicinal herbs have been explored as an alternative therapy in the management of DMT2 due to adverse effects of synthetic drugs used in allopathic medicine. $\alpha$-amylase and $\alpha$-glucosidase inhibitory potential and phytochemical profiling were investigated in aqueous extracts of two new *Cinnamomum zeylanicum* accessions, namely *C. zeylanicum* Sri Wijaya (SW), *C. zeylanicum* Sri Gemunu (SG) and commercially available *C. zeylanicum* (CC).

**Methods**. Microwave Digestion (MD), Pressurized Water Extraction (PWE), Steam Distillation (SD), Solvent Extraction (SE), Decoction Water Extraction (DWE) and Infusion Water Extraction (IWE) methods were used to prepare Cinnamon quill extracts. Total phenolic content (TPC, Folin-Ciocalteu method) and Proanthocyanidin content (PC, vanillin assay), $\alpha$-amylase and $\alpha$-glucosidase inhibition of Cinnamon extracts were determined spectrophotometrically. The $\alpha$-amylase and $\alpha$-glucosidase inhibition were reported in terms of $IC_{50}$ value. The phytochemical profiling was accomplished by GC-MS technique.

**Results and Discussion**. Lowest $IC_{50}$ values were observed in PWE and DWE of SW. The highest PC and TPC were also observed in PWE and DWE of SW. Pressured water and decoctions are promising methods for the extraction of antidiabetic constituents from cinnamon. Benzoic acid, cinnamyl alcohol, benzyl alcohol, and 4-Allyl-2,6-dimethoxyphenol were identified as major compounds in SW extracts. These compounds are believed to be responsible for strong enzyme inhibitory activity of the extracts.

**Conclusions**. This is the first study to explore the use of pressured and decoctions water to extract anti-diabetic phytochemicals from cinnamon. The extensive metabolite profiling of novel SW and SG extracts and comparison of that with commercially available CC are reported for the first time in this study. The *C. zeylanicum,* SW accession holds some promise in the management of diabetes.

# INTRODUCTION

Diabetes is a progressive metabolic disorder with multiple health effects and complications (*Schwartz et al., 2016*). More than 90% of diabetic patients are suffering from diabetes mellitus type 2 (DMT2). Modern sedentary lifestyle, unhealthy food habits and obesity are key factors responsible for the development of DMT2 (*William et al., 2019*). These factors may also contribute to generate a state of oxidative stress by producing reactive oxygen species. Oxidative stress can cause hyperglycemia by affecting the insulin secretion and its action *(Akash et al., 2011)*. Hence, the use of antioxidant based therapies in the treatment of DMT2 and its complications has been emphasized (*Rahimi-Madiseh et al., 2016*). Furthermore, there is a growing interest to develop herbal drugs and their bioactive compounds as an alternate therapy for DMT2 due to the serious side effects associated with synthetic therapeutic agents (*William et al., 2019*). One treatment strategy applicable for diabetes mellitus is the inhibition of enzymes involved in the hydrolysis of carbohydrates in the digestive tract. Number of bioactive ingredients in medicinal plants have been found to inhibit α-amylase and α-glucosidase enzymes (*Funke & Melzig, 2006*). α-amylase breaks down the large and insoluble starch molecules into absorbable disaccharides. α-glucosidase helps to convert oligosaccharides and disaccharides into monosaccharaides *(Kazeem, Ogunbiyi & Ashafa, 2013)*. The inhibitors of these enzymes restrict the digestion of dietary carbohydrates, thus subsequently prevent the absorption of simple sugars leading to low postprandial glucose levels in the blood. In addition, herbal extracts are also rich in natural antioxidants, which may help to reduce the risk of developing DMT2 (*Zaid et al., 2015*).

*Cinnamomum zeylanicum* (Family Lauraceae) which is commonly known as Ceylon cinnamon or true cinnamon is an indigenous plant in Sri Lanka. Recent studies have shown many beneficial health effects of Cinnamon such as anti-hypertensive effect, anti-inflammatory properties, anti-microbial activity, blood glucose control, reducing the risk of cardiovascular disease and colonic cancer (*Mahmoodnia, Aghadavod & Rafieian-Kopaei, 2017*; *Ouattara et al., 1997*; *Khan et al., 2003*; *Shen et al., 2010*).

Eight Cinnamon species, categorized on the basis of the taste of the bark, have been identified in Sri Lanka *(Azad et al., 2015)*. Among them only *Cinnamomum zeylanicum* is grown commercially. Recently, two accessions of *Cinnamomum zeylanicum* named as ''Sri Wijaya'' and ''Sri Gemunu'' have been developed for commercial cultivation with improved chemical profiles and yields (*Azad et al., 2015*). Yet, scientific evidence on biological activities and the anti-diabetic properties of these novel Cinnamon varieties have not been explored. This study was conducted with the objective of to evaluate six types of aqueous extracts of SW and SG cinnamon quill varieties and commercially available CC quills for their in vitro anti-diabetic activities and compare them with the standard hypoglycemic drug Acarbose. The study was further extended to explore the phytochemical profiles of different Ceylon cinnamon extracts.

## MATERIALS & METHODS

### Collection of plant materials and preparation of extracts

Dried commercial *Cinnamomum zeylanicum* quills (1 kg) were collected from the Dassanayake Walauwa Cinnamon plantation, Nape, Kosgoda, Southern Province in Sri Lanka. Sri Wijaya and Sri Gemunu Cinnamon quills (2 kg per each) were collected from Cinnamon Research Station, Palolpitiya, Thihagoda, Southern Province, Sri Lanka. The collected samples were transported in sealed, sterilized polythene bags to the laboratory at University of Kelaniya, Sri Lanka. The samples were stored in a refrigerator (2−8 °C) until use.

Cinnamon accessions were authenticated by a botanist at the Department of Botany, University of Kelaniya. The voucher specimen of "Sri Wijaya" and "Sri Gemunu" accessions were deposited at the publicly available herbarium, Department of Plant and Molecular Biology, the University of Kelaniya, Sri Lanka under the family Lauraceae (Deposition numbers are CIN-SW-001 and CIN-SG-002 respectively for "Sri Wijaya" and "Sri Gemunu" accessions).

Cinnamon quills (10 g) were pulverized using a 0.50 mm mesh. For the microwave digestion (MD), the sample (10 g) was digested with distilled water (80 mL) for 30 min using a microwave digester (mass 6 instrument, vessel type Mars Xpress). Pulverized Cinnamon quills (10 g) were extracted with pressurized water (200 mL at 0.098 MPa, for 10 min) for the preparation of pressurized water extract (PWE). Quills of Cinnamon (40 g) were extracted by traditional steam distillation (SD) and 10 g were extracted by solvent extraction (SE) using 75% ethanol (*Wong, 2014*; *Lee et al., 2018*). For the preparation of decoction water extract (DWE), Cinnamon quills (10 g) were boiled with water (200 mL) until the volume was reduced to 1/8. Ten grams of Cinnamon powder was mixed with boiled water (200 mL) and that was allowed to stand for five minutes to obtain the infusion water extract (IWE). The MD, PWE, DWE, and IWE were filtered through Whatman 1 filter paper and concentrated under vacuum at 45 °C and further dried by passing a stream of $N_2$ air. The percentage yield was calculated and stored at −20 °C. The volatile compounds from SD were separated from the aqueous layer three times using hexane (30 mL). The volatiles were concentrated by using rotary evaporator (IKA® RV 10 basic, Germany). The percentage yield of the resultant oleoresin was calculated and the oleoresin was stored at −20 °C.

### α-glucosidase inhibitory activity

The α-glucosidase inhibition assay (*Apostolidis & Lee, 2010*) was used to determine the in vitro anti-diabetic properties of Cinnamon extracts. Varying concentrations (12.5 µg/mL – 400 µg/mL) of Cinnamon quill extract (100 µL) and 0.1M phosphate buffer, pH 6.8 (100 µL) with α-glucosidase enzyme solution (1 Unit/mL) were incubated in 96 well plates at 37 °C for 10 min. After pre-incubation, 2.5 mM 4-Nitrophenyl β-D-glucopyranoside (pNPG) solution (20 µL) in 0.1M phosphate buffer, pH 6.8 was added to each well. The reaction mixture was incubated at 37 °C for 20 min. After the incubation, the absorbance at 405 nm was recorded by the micro plate reader (Spectra Max M5, Molecular Devices, CA,

USA). Acarbose was used as the positive control (12.5 μg/mL − 400 μg/mL). The solvent alone was used as the blank in the assay. The $IC_{50}$ values were calculated as follows;

$$\text{Inhibition (\%)} = 1 - \left\{ \frac{A_{sample}}{A_{control}} \right\} \times 100$$

where, $A_{sample}$ and $A_{control}$ were defined as absorbance of the sample and the control (blank) respectively.

## α- Amylase inhibitory activity

The α-amylase inhibition assay *(Ranilla et al., 2010)* was also used to evaluate the anti-diabetic properties of Cinnamon extracts. Various concentrations (12.5 μg/mL - 400 μg/mL) of the extract (100 μL) and 0.02 M sodium phosphate buffer, pH 6.9 (100 μL), amylase enzyme solution (0.5 mg/mL, 10 μL) were incubated at room temperature (28 ± 2 °C) for 10 min in a test tube. After pre-incubation, 100 μL of 1% starch in 0.02 M sodium phosphate buffer, pH 6.9 was added to each tube. The reaction mixtures were incubated at room temperature (28 ± 2 °C) for 10 min. The reaction was quenched by adding Dinitrosalicylic acid color reagent (100 μL). The test tubes were boiled in a water bath until the yellowish orange color was developed and then the tubes were allowed to cool. The reaction mixture was diluted with distilled water (5.00 mL), and a 250 μL aliquot of the reaction mixture was transferred into a 96 well micro titer plate. The absorbance at 540 nm was measured using a micro plate reader (Spectra Max M5, Molecular Devices, CA, USA). Acarbose was used as the positive control (12.5 μg/mL − 400 μg/mL).

The α-amylase inhibitory activity of each extract is expressed as percent inhibition which was calculated as follows:

$$\text{Inhibition (\%)} = 1 - \left\{ \frac{A_{sample}}{A_{control}} \right\} \times 100$$

where, $A_{sample}$ and $A_{control}$ are defined as absorbance of the sample and the control respectively. Control was conducted without adding the extract.

## Total phenolic content (TPC) determination assay

The Folin-Ciocalteu method *(Wang et al., 2012)* was used to determine the TPC. The plant extract (3.00 mg extract dry weight) was mixed with 10% Folin-Ciocalteu reagent (5.00 mL) and the mixture was incubated at room temperature (28 ± 2 °C) for five minutes. Sodium carbonate (4.00 mL, 7.5% v/w) was added and the mixture was allowed to stand for one hour at room temperature (28 ± 2 °C). The absorbance was measured at 765 nm using a UV-visible spectrophotometer. A calibration curve for Gallic acid in concentrations from 0.02 mg/mL to 1.00 mg/mL ($R^2 = 0.99$) was used to interpolate results of the TPC and the results were expressed as gallic acid equivalents (GAE) mg/g dried extract.

## Proanthocyanidine content (PC) determination assay

Vanillin assay was used to determine the PC. The extract (3.00 mg extract dry weight) of Cinnamon quills was mixed with 1% w/v vanillin solution in 7 M $H_2SO_4$ (4.00 mL) and they were incubated at room temperature (28 ± 2 °C) for 15 min. After the incubation, the absorbance was measured at 500 nm. Catechine was used as the standard. A calibration curve for catechin in concentrations from 0.05 mg/mL to 0.25 mg/mL ($R^2 = 0.99$) was used to interpolate the results of the PC and the results were expressed as catechin equivalents *(Toda, 2005)*.

## GC-MS analysis

Crude extract (4.00 mg) was dissolved in 1.00 mL of hexane and the samples were filtered using a nylon filter (0.45 µm pore size). A 0.7 µL aliquot of the above was injected in the split less mode into a GC/MS 7890B Gas Chromatograph (Agilent, American) equipped with a 5977B mass spectrometer (Agilent, American). A fused silica capillary Agilent Technology HP-5 (5% phenyl-methyl polysiloxane) column (30 m × 0.25 mm × 0.25 µm) was used for the separation. The injector temperature was 250 °C. The initial temperature was kept at 40 °C, for 3 min and the temperature was gradually increased to 220 °C at the rate of 4 °C min$^{-1}$ and was then held for 2 min at 220 °C. Again the temperature was gradually increased to 230 °C at the rate of 8 °C min$^{-1}$ and held for 3 min at 230 °C. The Post run was at 235 °C held for 3 min. The Total GC running time was 54.25 min. Helium was used as the carrier gas at a constant flow rate of 1.0 mL min$^{-1}$ at the split-less mode. EI was used as the ion source, and the ion source temperature was 230 °C. The sector mass analyzer was set to scan from 40 to 650 amu The volatile components of the extracts were identified using mass spectral data, with computer assisted matching with WILEY 275 and National Institute of Standards and Technology (NIST6.0) libraries.

## Statistical analysis

All results are expressed as mean ± SD of triplicate assays. Statistical analyses were performed using Microsoft Office Excel (2013) and Graph Pad Prism 7 statistical package (GraphPad Software, USA). Significant differences among the data were analyzed by the SPSS Statistical computer package (IBM® SPSS® Statistics Version 23, USA). The results were analyzed using one-way ANOVA test followed by Dunnett/Tukey test for multiple comparisons, paired sample $t$-test and determination of significance level. Group means were considered to be significantly different at $P < 0.05$.

# RESULTS

## Extract yield analysis

The yield of Cinnamon quill extracts ranged from 0.42% ± 0.03% to 2.19% ± 0.25% for SW; 0.39% ± 0.02% to 1.90% ± 0.05% for SG and 0.15% ± 0.00% to 1.26% ± 0.08% for CC (Fig. 1). The results indicated that there was significant difference ($p < 0.05$) in the yield of the extracts depending on the Cinnamon accessions and the extraction methods. The highest yield was obtained from SW, PWE (2.19% ± 0.25%). Microwave digestion gave the lowest yield in all Cinnamon varieties (0.42% ± 0.03% for SW; 0.39% ± 0.02% for SG and 0.15% ± 0.00% for CC).

## α-glucosidase and α-amylase enzyme inhibitory activity

The α-glucosidase and α-amylase enzyme inhibitory activities of cinnamon extracts are expressed in terms of IC$_{50}$ values, and the lower IC$_{50}$ is an indicative of a stronger inhibition. SW, PWE exhibited the highest inhibitory activity ($P < 0.05$) for α-glucosidase and α-amylase (42 ± 8 µg mL$^{-1}$ and 78 ± 7 µg mL$^{-1}$, respectively) followed by SW, DWE (116 ± 17 µg mL$^{-1}$ and 132 ± 11 µg mL$^{-1}$, respectively). The inhibition of these extracts were comparable with the positive control, Acarbose (173 ± 7 µg mL$^{-1}$ and 95 ± 4 µg mL$^{-1}$ respectively) (Table 1).
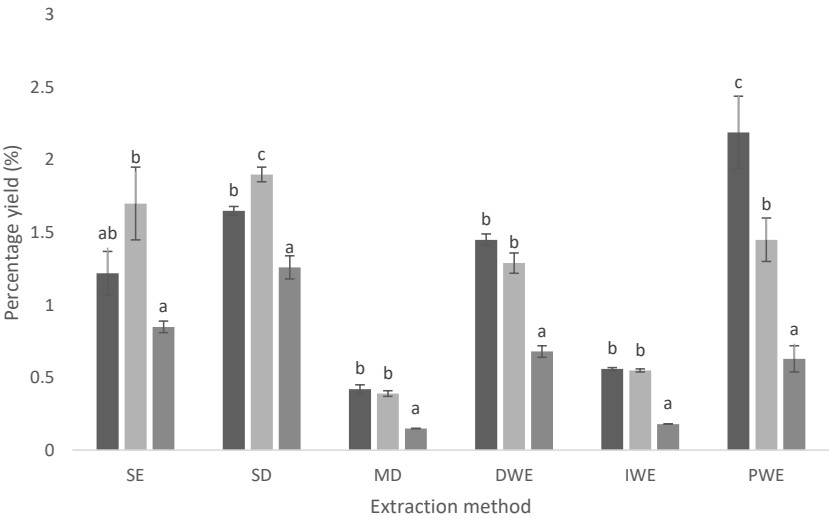

**Figure 1** **Yields (%, w/w, dry basis) of compounds in extracts from different cinnamon quills with different extraction methods ($n = 3$).** Mean values in each extraction methods, in each cinnamon varieties followed by the same letter, are not significantly different ($p \geq 0.05$) by Tukey's multiple range comparison tests. Black: SW, Sri Wijaya cinnamon accession; Whitish gray: SG, Sri Gemunu cinnamon accession; Gray: CC, commercially available *Cinnamomum zeylanicum*; SE, Solvent Extraction; SD, Steam Distillation; MD, Microwave Digestion; DWE, Decoction Water Extraction; IWE, Infusion Water Extraction; PWE, Pressurized Water Extraction.

## TPC and PC analysis

Total phenolic content was expressed as mg GAE/g and PC was expressed as mg of catechin equivalent/g. In comparing the TPC and the PC of extracts, the highest TPC and the highest PC were obtained for the DWE ($2.24 \pm 0.00$ mg GAE/g and $40.05 \pm 0.10$ mg of catechin equivalent/g respectively) for SW. The extracts obtained by SD had the lowest TPC and PC for SW ($0.21 \pm 0.01$ mg GAE/g and $1 \pm 0.01$ mg of catechin equivalent/g) (Table 2).

## GC-MS analysis

(*E*)- Cinnamaldehyde, was the predominant compound in CC, SW and SG extracts prepared using steam distillation, Infusion and solvent extraction respectively (Table 3). In pressured water extracts of SW and SG; Benzoic acid (1.6%, 15.49%), (*E*)-Cinnamaldehyde (5.8%, 5.46%), Cinnamyl alcohol (32.44%, 40.08%), and 4-Allyl-2,6-dimethoxyphenol (10.2%, 12.71%) were the major compounds present (Table 3).

Benzoic acid (2.58%, 22.51%), (*E*)-Cinnamaldehyde (34.6%, 5.58%), Trans-cinnamic acid (16.86%, 4.51%), O-Methoxy-cinnamaldehyde (2.04%, 2.14%), Cinamyl alcohol (21.3%, 42.48%), and 4-Allyl-2,6-dimethoxyphenol (3.24%, 8.79%) were the major compounds present in SG and SW decoction water extracts. Benzoic acid and Cinnamyl alcohol were not detected in CC, PWE and DWE. 4-Allyl-2,6-dimethoxyphenol (1.79%, 1.05%) was seen as a minor constituent in CC extracted using pressured water and as a decoction (Table 3).

Twenty seven chemical components were identified for the first time in SW; PWE. Benzcatechin (1.14%), 1-Methoxy-7-methyl-3,4-dihydrobenzo[c]pyran (1.31%),

**Table 1  $IC_{50}$ for $\alpha$-glucosidase and $\alpha$-amylase of SW, SG and CC, extracted by different methods.** Each data point represents the mean of three replicates $\pm$ SEM. Mean values in each column, in each cinnamon varieties followed by the same letter, are not significantly different ($p \geq 0.05$) by Tukeys multiple range comparison tests.[a]

| Extraction methods | $IC_{50}$ for $\alpha$-glucosidase ($\mu g\,mL^{-1}$) | $IC_{50}$ for $\alpha$-amylase ($\mu g\,mL^{-1}$) |
|---|---|---|
| "SW" | | |
| SE | $650 \pm 121^c$ | $615 \pm 99^c$ |
| SD | $150 \pm 5^{ab}$ | $172 \pm 9^b$ |
| MD | $160 \pm 14^b$ | $192 \pm 7^b$ |
| DWE | $116 \pm 17^{ab}$ | $132 \pm 11^{ab}$ |
| IWE | $144 \pm 31^{ab}$ | $171 \pm 14^b$ |
| PWE | $42 \pm 13^a$ | $78 \pm 7^a$ |
| Acarbose | $173 \pm 7^b$ | $95 \pm 4^a$ |
| "SG" | | |
| SE | $785 \pm 124^c$ | $708 \pm 77^d$ |
| SD | $165 \pm 5^{ab}$ | $180 \pm 7^c$ |
| MD | $182 \pm 6^b$ | $215 \pm 12^c$ |
| DWE | $121 \pm 5^a$ | $143 \pm 16^{bc}$ |
| IWE | $149 \pm 2^a$ | $169 \pm 17^c$ |
| PWE | $72 \pm 3^{ab}$ | $186 \pm 10^b$ |
| Acarbose | $173 \pm 7^b$ | $95 \pm 4^a$ |
| "CC" | | |
| SE | $606 \pm 139^e$ | $201 \pm 5^c$ |
| SD | $296 \pm 25^d$ | $120 \pm 5^b$ |
| MD | $159 \pm 6^b$ | $111 \pm 2^a$ |
| DWE | $162 \pm 8^b$ | $260 \pm 13^d$ |
| IWE | $129 \pm 9^a$ | $246 \pm 17^d$ |
| PWE | $132 \pm 5^a$ | $88 \pm 7^a$ |
| Acarbose | $173 \pm 7^b$ | $95 \pm 4^a$ |

**Notes.**

[a]SW, Sri Wijaya cinnamon accession; SG, Sri Gemunu cinnamon accession; CC, commercially available *Cinnamomum zeylanicum*; SE, Solvent Extraction; SD, Steam Distillation; MD, Microwave Digestion; DWE, Decoction Water Extraction; IWE, Infusion Water Extraction; PWE, Pressurized Water Extraction.

(+)-(1S,3R,4S)-4(a)-Methyladamantane (2.14%), 1-(2′-hydroxyphenyl)prop-2-en-1-ol (0.71%), 1,3,5-benzenetriol (1.11%), 4-propyl-1,2-Benzenediol (1.615%), 3,4,5-trimethoxyphenol (0.54%), 4-((1E)-3-Hydroxy-1-propenyl)-2-methoxyphenol (1.00%), and Vanillylmandelic acid (0.78%) were present only in PWE of SW. These compounds and 4-Allyl-2,6-dimethoxyphenol were detected for the first time in any Cinnamon accessions (Table 3).

## DISCUSSION

Cinnamon incorporated foods and nutraceuticals are popular choice in Sri Lanka due to the belief of its anti-diabetic properties. However, the anti-diabetic effect of two selected cinnamon accessions, *C. zeylanicum* Sri Wijaya and *C. zeylanicum* Sri Gemunu have not

**Table 2  TPC and PC from quills of CC, SW and SG, with different extraction methods.** Each data point represents the mean of three replicates $\pm$ SD. Mean values in each column, in each cinnamon accession followed by the same letter, are not significantly different ($p \geq 0.05$) by Tukey's multiple range comparison tests.[a]

| Extraction methods | TPC (mg GAE/g) | | | PC (mg catechin equivalent/g) | | |
|---|---|---|---|---|---|---|
| | "SW" | "SG" | "CC" | "SW" | "SG" | "CC" |
| SE | $0.90 \pm 0.01^e$ | $0.65 \pm 0.01^d$ | $0.90 \pm 0.01^b$ | $5.12 \pm 0.01^b$ | $3.56 \pm 0.01^b$ | $7.34 \pm 0.01^c$ |
| SD | $0.21 \pm 0.01^a$ | $0.15 \pm 0.07^b$ | $0.68 \pm 0.01^a$ | $3.00 \pm 0.01^a$ | $1.00 \pm 0.01^a$ | $5.65 \pm 0.01^a$ |
| MD | $0.67 \pm 0.01^c$ | $0.12 \pm 0.01^a$ | $1.73 \pm 0.02^d$ | $15.00 \pm 0.01^c$ | $12.00 \pm 0.01^c$ | $6.14 \pm 0.01^b$ |
| DWE | $2.24 \pm 0.00^f$ | $1.00 \pm 0.00^f$ | $0.91 \pm 0.01^b$ | $40.05 \pm 0.10^f$ | $26.00 \pm 0.09^f$ | $17.5 \pm 0.01^d$ |
| IWE | $0.87 \pm 0.01^d$ | $0.83 \pm 0.01^e$ | $1.51 \pm 0.05^c$ | $27.21 \pm 0.07^d$ | $21.78 \pm 0.09^e$ | $32.77 \pm 0.01^e$ |
| PWE | $1.53 \pm 0.01^b$ | $0.18 \pm 0.01^c$ | $2.90 \pm 0.08^e$ | $36.08 \pm 0.01^e$ | $17.21 \pm 0.01^d$ | $7.39 \pm 0.03^c$ |

**Notes.**
[a]SW, Sri Wijaya cinnamon accession; SG, Sri Gemunu cinnamon accession; CC, commercially available *Cinnamomum zeylanicum*; SE, Solvent Extraction; SD, Steam Distillation; MD, Microwave Digestion; DWE, Decoction Water Extraction; IWE, Infusion Water Extraction; PWE, Pressurized Water Extraction.

been scientifically verified and not well understood. In the current study, water extracts were prepared from selected Ceylon Cinnamon accessions to determine the anti-diabetic activity. The α-glucosidase/α-amylase inhibitory activities, total phenolic content, and proanthocyanidin content were evaluated in aqueous extracts of *C. zeylanicum* and the chemical profiles were identified using GC-MS.

The yield of extracts prepared using different methods from variety of Cinnamon quill samples are presented in Fig. 1. The yield of the bark essential oil of novel Cinnamon accessions prepared using hydro distillation have been reported previously for SW (1.25%), SG (1.49%) (*Ariyarathne, Weerasuriya & Senarath, 2018*), and CC (1.2%) (*Jayawardena & Smith, 2010*). Higher yield was reported for Sri Gemunu Cinnamon accessions (3.4%. v/w) than for the Sri Wijaya by the same distillation method in another study (*Lokuge et al., 2018*). In the current study, the percent yields obtained for SW, SG, and CC extracts prepared using SD were comparable to the previously reported yield (2.6%) for the commercially available *C. zeylanicum* in Turkey (*Sihoglu Tepe & Ozaslan, 2020*). Another finding showed a 352 mg/g yield for the freeze dried aqueous extract of C. *zeylanicum* and that was higher than the yield obtained in the current study (*Takács et al., 2017*). Species of Cinnamon, climate, growth condition, cultivation site, age of the bark, thickness of the bark and the density of the oil cells are positively correlated with the yield of the Cinnamon essential oils (*Li, Kong & Wu, 2013*). Color of the bark, peeling ability, texture of the bark, odor and the morphology of the leaves have been examined in SW and SG varieties (*Ariyarathne, Weerasuriya & Senarath, 2018*). However, scientific evidences on the bark thickness, density of the oil cells of SW and SG have not been explored. Hence further investigations need to be done to evaluate the properties of the bark of these accessions.

The inhibitory activity for α-glucosidase and α-amylase varied depending on the sample and the type of extraction. By comparing the α-glucosidase and α-amylase inhibitory activities of the samples tested, the SW, PWE exhibited the strongest inhibition compared to the positive control, Acarbose. The methanolic extracts of Ceylon Cinnamon have been reported to have the potential to control hyperglycemia (*Nair, Kavrekar & Mishra, 2013*).

Wariyapperuma et al. (2020), *PeerJ*, DOI 10.7717/peerj.10070

**Table 3 Chemical compositions of extracts from CC. SW and SG, by various extraction methods.** Each data point represents the mean of three replicates.

| Compounds | SG | | | | | CC<br>Area percentage (%) | | | | | SW | | | | |
|---|---|---|---|---|---|---|---|---|---|---|---|---|---|---|---|
| | 1 | 2 | 3 | 4 | 5 | 1 | 2 | 3 | 4 | 5 | 1 | 2 | 3 | 4 | 5 |
| 3-carene | – | – | – | – | – | 0.2 | – | – | – | – | – | – | – | – | – |
| Cymol | – | – | – | 0.89 | – | – | – | – | – | – | – | – | 0.16 | – | – |
| Benzoic acid | – | – | 1.6 | 2.58 | – | – | – | – | – | – | – | – | 15.49 | 22.51 | 0.81 |
| Benzyl alcohol | – | – | 0.17 | – | – | – | – | 0.42 | – | – | – | – | 3.8 | 3.98 | – |
| Undecane | – | – | 0.46 | – | – | | | | | | – | 1.1 | – | – | |
| Methyl salicylate | – | – | 0.26 | – | 2.34 | | – | 13.52 | 10.35 | – | | – | 0.42 | – | 5.24 |
| Terpinen-4-ol | – | 1.36 | – | – | – | 0.48 | 0.41 | – | – | – | 0.78 | 0.44 | – | – | – |
| Naphthalene | – | – | – | – | – | 0.19 | – | – | – | – | – | – | – | – | – |
| Linalool | – | 5.17 | – | – | – | 2.6 | 2.07 | – | – | 0.31 | 1.07 | 2.21 | – | – | – |
| Benzcatechin | – | | | | | | | | | | – | 1.14 | – | – | |
| β-Fenchyl alcohol | – | | | | | | 0.77 | – | | 0.31 | | 0.81 | – | – | – |
| Benzenepropanal | – | 1.68 | – | – | – | 0.14 | 0.41 | – | 0.35 | 0.24 | 0.14 | 0.43 | – | – | 0.34 |
| Benzenepropanol | – | – | 1.2 | 2 | 0.74 | | – | 0.29 | | 0.46 | 0.19 | – | 0.54 | – | 0.34 |
| 3-cyclohexene-1-methanol | – | – | – | – | – | 0.69 | – | – | – | – | – | – | – | – | – |
| Salicylic acid | – | – | – | – | – | – | – | 12.93 | 4.05 | – | – | – | – | – | – |
| Acetic acid, cinnamyl ester | – | – | – | – | – | – | 6.05 | 1.74 | 1.85 | – | – | 2.85 | – | – | – |
| (E)-cinnamaldehyde | 43 | 65.21 | 5.8 | 34.6 | 55.72 | 48.77 | 79.06 | 43.41 | 57.7 | 73.25 | 39.24 | 38.85 | 5.46 | 5.58 | 40.48 |
| Benzaldehyde | – | – | – | – | – | – | – | – | – | – | – | 0.07 | – | – | – |
| 1-phellandrene | – | – | – | – | – | 1.28 | – | – | – | – | – | 0.09 | – | – | – |
| (E)- 2,3-epoxycarane | – | – | – | 0.83 | – | – | – | – | – | – | – | – | – | – | – |
| Dimethylsulfonium dicyanomethylide | – | – | – | 0.4 | – | – | – | – | – | – | – | – | – | – | – |
| 2,4-Dimethyl-2,4-pentadien-1-ol | – | – | – | 1.58 | – | – | – | – | – | – | – | – | – | – | – |
| Eugenol | 8 | 2.47 | 0.33 | 0.59 | 1.82 | 1.94 | 6.75 | 7.85 | 7.67 | 8.83 | 14.67 | 12.51 | 1.74 | 1.27 | 6.55 |
| Cinnamaldehyde dimethyl acetal | – | – | 11.84 | 5.3 | 18.53 | 6.2 | – | 0.54 | 0.32 | 2.06 | 0.49 | – | 0.39 | – | 5.74 |
| Vanillylmandelic acid | – | – | 0.78 | – | – | – | – | – | – | – | – | – | 0.78 | – | – |
| Trans-Cinnamic acid | – | – | 0.14 | 16.86 | 2.07 | – | – | – | – | – | – | – | 1.74 | 4.51 | 0.92 |
| O-methoxy-Cinnamaldehyde | – | 0.65 | 0.95 | 2.04 | 1.78 | 1.97 | 1.79 | – | 1.52 | 2 | 2.62 | 2.18 | 2.04 | 2.14 | 3.57 |
| Para methoxy cinnamic aldehyde | – | | | | | | – | 1.66 | 1.85 | 0.36 | – | – | – | – | – |
| Propanoic acid, phenylmethyl ester | – | – | – | 0.4 | – | – | – | – | – | – | – | – | – | – | – |
| 1,2-Dihydroxy-4-(1-propyl)benzene | – | – | – | 0.86 | – | – | – | – | – | – | – | – | – | – | – |
| Benzyl benzoate | 20 | 1.17 | – | 0.6 | 0.39 | 3.42 | 2.69 | 0.33 | 1.52 | 0.62 | 5.78 | 6.29 | – | 1.15 | 1.74 |

**Table 3** (*continued*)

| Compounds | SG | | | | | CC Area percentage (%) | | | | | SW | | | | |
|---|---|---|---|---|---|---|---|---|---|---|---|---|---|---|---|
| | 1 | 2 | 3 | 4 | 5 | 1 | 2 | 3 | 4 | 5 | 1 | 2 | 3 | 4 | 5 |
| Sabinene | – | 0.97 | – | – | – | – | – | – | – | – | – | 0.1 | – | – | – |
| γ-terpinene | – | – | – | 0.42 | – | 0.24 | – | – | – | – | – | 0.11 | – | – | – |
| Cinnamyl alcohol | 1 | 0.72 | 32.44 | 21.3 | 10.87 | 0.7 | – | – | – | 6.15 | 5.55 | 0.66 | 40.08 | 42.28 | 14.32 |
| Phenylpropyl acetate, | – | – | – | – | – | – | – | – | – | – | 0.61 | – | – | – | – |
| Copaene | – | – | – | – | – | 0.46 | – | – | – | – | – | – | – | – | – |
| Caryophyllenyl alcohol | – | – | – | – | – | 0.33 | – | – | – | – | – | – | – | – | – |
| (+) Spathulenol | – | – | – | – | – | 0.24 | – | – | – | – | – | – | – | – | – |
| Caryophyllene | 3 | – | – | 0.51 | – | 4.33 | – | – | – | – | 0.26 | – | – | – | – |
| trans-Cinnamyl acetate | 23 | 19.85 | – | – | 3.52 | 3.85 | – | – | – | 2.09 | 25.39 | 28.59 | 0.13 | – | 9.73 |
| (+)-(1S,3R,4S)-4(a)-Methyladamantane | – | – | – | – | – | – | – | – | – | – | – | 2.14 | – | – | – |
| 1-(2′-hydroxyphenyl)prop-2-en-1-ol | – | – | – | – | – | – | – | – | – | – | – | 0.71 | – | – | – |
| (1S,2S,6S,8S)-11-(Hydroxymethyl)-6-methyl-3 methylenetricyclo[6.3.0.0(2,6)]undec | – | – | – | – | – | – | – | – | – | – | – | 0.1 | – | – | – |
| 1,3,5-benzenetriol | – | – | – | – | – | – | – | – | – | – | – | 1.11 | – | – | – |
| 4-propyl-1,2-Benzenediol | – | – | – | – | – | – | – | – | – | – | – | 1.61 | – | – | – |
| Benzenemethanol | – | – | – | – | – | – | – | – | 0.18 | 0.17 | – | – | – | 0.08 | |
| Caryophyllene oxide | – | – | – | – | – | 2.18 | – | – | – | – | – | 0.11 | – | – | – |
| Allylbenzene | – | – | – | – | – | – | – | – | – | – | – | – | – | 0.18 | |
| 3-Phenylprop-2-yn-1-ol | – | – | – | – | – | – | – | – | – | 0.2 | – | – | – | – | – |
| 1-(Trideuteriosylanyl)-benzene | – | – | – | – | – | – | – | – | – | 0.16 | – | – | – | – | – |
| Methoxy-phenyl-Oxime | – | – | – | – | – | – | – | – | – | – | – | 0.04 | – | – | – |
| Cineole | – | – | – | – | – | – | – | – | – | – | – | 0.04 | – | – | – |
| 3,7-dimethyl- 1,6-octadien-3-ol | – | – | – | – | – | – | – | – | – | – | – | 0.26 | – | – | – |
| α-terpinolene | – | – | – | – | – | – | – | – | – | – | – | 0.41 | – | – | 0.07 |
| O-Methoxyphenol | – | – | – | – | – | – | – | – | – | – | – | 0.1 | – | – | – |
| Vinyl phenyl carbinol | – | – | – | – | – | – | – | – | – | 0.08 | 0.16 | – | – | – | |
| Ethyl benzenecarboxylate | – | – | – | – | – | – | – | – | – | – | – | 0.07 | – | – | – |
| 2H-1-benzopyran | – | – | – | – | – | – | – | – | – | – | – | 0.06 | – | – | – |
**Table 3** (*continued*)

| Compounds | SG | | | | | CC Area percentage (%) | | | | | SW | | | | |
|---|---|---|---|---|---|---|---|---|---|---|---|---|---|---|---|
| | 1 | 2 | 3 | 4 | 5 | 1 | 2 | 3 | 4 | 5 | 1 | 2 | 3 | 4 | 5 |
| 2-methoxy-4-propyl-Phenol | – | – | – | – | – | – | – | – | – | – | – | 0.12 | – | – | – |
| Benzenepropyl acetate | – | – | – | – | – | – | – | – | – | – | – | 0.79 | – | – | – |
| Homo - syringaldehyde | – | – | – | – | – | – | – | – | – | – | – | 0.04 | – | – | – |
| Ortho methoxy cinnamyl acetate | – | – | – | – | – | – | – | – | – | – | 0.36 | 0.43 | – | – | 0.68 |
| Oxalic acid, 2-phenylethyl propyl ester | – | – | – | – | – | – | – | – | – | – | – | 0.21 | – | – | – |
| Benzyl salicylate | – | – | – | – | – | – | – | – | – | – | – | 0.03 | – | – | – |
| Methyl palmitate | – | – | – | – | – | – | – | – | – | – | – | 0.06 | – | – | – |
| Linoleic acid | 1 | – | – | – | – | 0.83 | – | – | – | – | 0.13 | 0.03 | – | – | – |
| 3,4,5-trimethoxyphenol | – | – | – | – | – | – | – | – | – | – | – | – | 0.54 | – | – |
| Benzaldehyde, 4-hydroxy-3,5-dimethoxy- | – | – | – | – | – | – | – | – | – | – | – | – | 0.28 | – | – |
| 3-Methoxy-4-hydroxycinnamaldehyde | – | – | – | – | – | – | – | – | – | – | – | – | 0.43 | – | – |
| 4-((1E)-3-Hydroxy-1-propenyl)-2-methoxyphenol | – | – | – | – | – | – | – | – | – | – | – | – | 1.00 | – | – |
| 3-Phenylprop-2-yn-1-ol | – | – | – | – | – | – | – | – | – | – | – | – | – | – | 0.22 |
| Salicylic acid | – | – | – | – | – | – | – | – | – | – | – | – | – | – | 2.95 |
| Formic acid, 3-phenylpropyl ester | – | – | – | – | – | – | – | – | – | – | – | – | – | – | 0.08 |
| 4,2,8-Ethanylylidene-2H-1-benzopyran, octahydro-2-methyl- | – | – | – | – | – | – | – | – | – | – | – | – | – | – | 0.13 |
| 1-Methoxy-7-methyl-3,4-dihydrobenzo[c]pyran | – | – | – | – | – | – | – | – | – | – | – | – | – | – | 0.4 |
| 2,6-dimethoxy-4-(2-propenyl)-Phenol, | – | – | – | – | – | – | – | – | – | – | 0.98 | | | | |
| Butylated hydroxytoluene | – | – | – | – | – | – | – | – | – | 0.4 | – | – | – | 1.85 | 0.21 |
| Trans-3-Pinen-2-ol [2,6,6-trimethylbicyclo[3.1.1]hept-3-en-2-ol] | – | – | – | – | – | – | – | – | – | 0.49 | | | | | |
| 4-Allyl-2,6-dimethoxyphenol | – | – | 10.2 | 3.24 | 0.92 | – | – | 1.79 | 1.05 | 1.7 | – | 0.77 | 12.71 | 8.79 | 2.97 |
| 4,7-Dihydro-4,7-methano-2H-indole | – | – | – | – | – | – | – | 0.5 | | – | | | | | |
| Palmitic acid | – | – | – | 0.44 | – | 1.53 | – | – | – | – | 0.28 | 0.05 | – | – | – |
| 4,4,8-Trimethyltricyclo [6.3.1.0(1,5)] dodecane-2,9-diol | – | – | – | 0.8 | – | – | – | – | – | – | – | – | 0.27 | – | – |

| Compounds | SG | | | | | CC<br>Area percentage (%) | | | | | SW | | | | |
|---|---|---|---|---|---|---|---|---|---|---|---|---|---|---|---|
| | 1 | 2 | 3 | 4 | 5 | 1 | 2 | 3 | 4 | 5 | 1 | 2 | 3 | 4 | 5 |
| 3-Methoxy-4-hydroxycinnamaldehyde | – | – | – | 0.31 | – | – | – | – | – | – | – | – | – | – | – |
| Styrene | – | – | 0.09 | – | – | – | – | – | – | – | – | – | – | – | – |
| Camphene | – | – | 0.08 | – | – | – | – | – | – | – | – | – | – | – | – |
| 1-Methyl-2-isopropylbenzene | – | – | 0.81 | – | – | – | – | – | – | – | – | – | – | – | – |
| Dodecane | – | – | 0.15 | – | – | – | – | – | – | – | – | – | – | – | – |
| 3,4,4-Trimethyl-2-pentenal | – | – | 0.38 | – | – | – | – | – | – | – | – | – | – | – | – |
| 2-methylene-Cyclohexanol | – | – | 1.29 | – | – | – | – | – | – | – | – | – | – | – | – |
| 1,5,9,9-tetramethyl-,Z,Z,Z-1,4,7,-Cycloundecatriene, | – | – | – | – | – | 1.15 | – | – | – | – | – | – | – | – | – |
| Δ-Cadinene | – | – | – | – | – | 0.23 | – | – | – | – | – | – | – | – | – |
| Hydrocinnamic acid | – | – | 0.2 | – | – | 0.42 | – | – | – | – | – | – | – | – | – |
| Phenol, 2,6-bis(1,1-dimethylethyl)- | – | – | 0.09 | – | – | – | – | – | – | – | – | – | – | – | – |
| (S)-(+)-5-sec-Butyl-2-pyrimidinol | – | – | 0.56 | – | – | – | – | 14.08 | 10.25 | – | – | – | – | – | – |
| Spiro[2-ethylidene-3-methylcyclohexane]oxirane | – | – | 1.96 | – | – | – | – | – | – | – | – | – | – | – | – |
| 1-Methoxy-7-methyl-3,4-dihydrobenzo[c]pyran | – | – | – | – | 0.28 | – | – | – | – | – | – | – | 1.31 | – | 1.3 |
| Aspirin methyl ester | – | – | – | – | 0.39 | – | – | – | – | – | – | – | – | – | 0.09 |
| 1H-Pyrrole-2,4-dicarboxylic acid, 3,5-dimethyl-, di-ethyl ester | – | – | – | – | 0.4 | – | – | 0.94 | 0.85 | – | – | – | – | – | 1.31 |
| Hexadecanoic acid | 1 | | | | | | | | | | | | | | |
| α-Thujene | | | | | | 0.27 | | | | | | | | | |

**Notes.**

SW, Sri Wijaya cinnamon accession; SG, Sri Gemunu cinnamon accession; CC, commercially available *Cinnamomum zeylanicum*; 1, solvent extraction; 2, steam distillation; 3, pressurized water extraction; 4, decoction water extraction; 5, infusion water extraction.

Wariyapperuma et al. (2020), *PeerJ*, DOI 10.7717/peerj.10070

The IC$_{50}$ values for the α-amylase and α-glucosidase inhibition were 130.55 ± 10.50 μg mL$^{-1}$ and 140.01 ± 10.08 μg mL$^{-1}$, respectively for the *C. zeylanicum* methanol extracts. Comparable α-amylase inhibition has been reported for methanol extract of Cinnamon bark (IC$_{50}$: 86.84 μg mL$^{-1}$), Cinnamon stick (IC$_{50}$: 54.69 μg mL$^{-1}$) and the clinical drug, Metformin hydrochloride (53.03 μg mL$^{-1}$) (*Wickramasinghe, Peiris & Padumadas, 2018*). According to the findings of *Wickramasinghe, Peiris & Padumadas (2018)*, a poor α-amylase inhibition was seen in Cinnamon drink (3207.01 μg mL$^{-1}$), Cinnamon capsule (2537.49 μg mL$^{-1}$), and Cinnamon powder (771.67 μg mL$^{-1}$). When compared with previously reported values for methanolic extracts, the PWE and DWE of SW showed better enzyme inhibition (IC$_{50}$ for the α-glucosidase inhibition; 42 ± 8 μg mL$^{-1}$ and 116 ± 17 μg mL$^{-1}$ and IC$_{50}$ for the α-amylase inhibition; 78 ± 7 μg mL$^{-1}$ and 132 ± 11 μg mL$^{-1}$ respectively). The enzyme inhibition mechanism of Acarbose is well established. Acarbose inhibit the α-glucosidase activity as a competitive inhibitor (*Van de Laar, 2008*). Phytochemicals in plant extracts, in particular phenolics act as non-competitive inhibitors against digestive enzymes. Non-competitive mode of inhibition is better due to the possible multiple side interactions of phenolic compounds with enzyme molecule and also this type of inhibition does not depend on the substrate concentration.

Polyphenols, in particular proanthocyanidin polymers are a large groups of phytochemicals that can be extracted from plants such as tea, coffee, wine, cocoa, grains, legumes, fruits and berries (*Chen et al., 2012*). Several findings have confirmed that there is a positive correlation between the antioxidant such as polyphenols present and the α-glucosidase and α-amylase inhibition (*Cai et al., 2004*; *Peng et al., 2010*; *Abeysekera, Premakumara & Ratnasooriya, 2013*; *Chandrasekharan & Bentota, 2013*). SW, DWE and PWE had high contents of PC and TPC, which can be correlated with the observed better anti-diabetic activities (Table 2).

The effectiveness of phenolic compounds in inhibiting α-amylase depends on the number and the position of hydroxyl groups (*Funke & Melzig, 2006*). In the current study, a high amount of phenolic compounds were extracted when water was used as a solvent and hence the extracts had potent anti-diabetic properties.

GC–MS technique is a powerful and suitable tool for the determination of volatile compounds because of its high separation efficiency and sensitive detection (*Li, Kong & Wu, 2013*). Cinnamaldehyde and Cinnamic acid are the major compounds of Cinnamon aqueous extracts (*Hafizur et al., 2015*). The chemical composition in the aqueous extracts in our study, slightly deviated from *Hafizur et al. (2015)* findings. Benzoic acid, Cinnamyl alcohol, Trans-Cinnamic acid and 4-Allyl-2,6-dimethoxyphenol were the major compounds present in SW and SG aqueous extracts. According to the findings of *Hafizur et al. (2015)*, it is desirable to have high concentrations of Cinnamic acid as it has the potential to decrease the blood glucose levels, improve glucose tolerance and stimulate insulin secretion in diabetic rats in a time and dose dependent manner. Hence, it is favorable to have extracts with high concentrations of Cinnamic acid for better diabetic control. Cinnamaldehyde is identified as the major compound present in Cinnamon extracted using hydro distillation in several studies (*Kaskoos, 2019*; *Mota, Campelo & Frota, 2019*). The α-amylase inhibition evaluated using a combination of (*E*)-Cinnamaldehyde

and (*E*)-Cinnamyl acetate has shown better potency than Cinnamaldehyde alone (*Sihoglu Tepe & Ozaslan, 2020*). The synergy between (*E*)-Cinnamaldehyde and (*E*)-Cinnamyl acetate seems to be important for potent enzyme inhibition. Therefore, poor enzyme inhibition observed in SD and SE are consistent with the absence of (*E*)-Cinnamyl acetate. Higher amount of (*E*)-Cinnamaldehyde alone does not improve the enzyme inhibitory potential. The inhibition by PWE of SW accession was higher due to the synergistic effect of the compounds present in the extract.

Preparation of decoction is a common method practiced in the preparation of Ayurveda drugs in therapeutic regimen (*Daswani et al., 2011*). Pressurize water is an environmentally friendly non-toxic novel method for effective extraction of plant metabolites without using organic solvents. However, the most commonly practiced method for the extraction of Cinnamon is steam distillation or hydro distillation. The loss of volatile compounds and long extraction times are some of the drawbacks of steam distillation.

Super critical Carbon dioxide is an alternative used for the extraction of thermo sensitive phytochemicals. However, in some studies supercritical carbon dioxide extracts have exhibited low antioxidant activity compared to the ethanol extracts due to the low polarity of super critical carbon dioxide as a solvent compared with ethanol (*Singh et al., 2007*).

Hence, there is an interest to develop new methods of extraction which could give better yields and biological activity. Pressurized water extraction has the advantage of extracting polar compounds from plant extracts hence can impart better biological activity (*Jayawardena & Smith, 2010*). In the current study, the extract prepared using pressured water had the most potent anti-diabetic activities. Further, low cost, safe for human consumption, short extraction times compared to steam distillation are added advantages of pressurized water extractions.

## CONCLUSIONS

Whilst there are numerous studies on the biological activity of essential oils, aspects such as safety, astringency, cost and the solubility in water stifle their applications as nutraceuticals. This study was mainly focused on investigating the in vitro hypoglycemic effects of aqueous extracts of Cinnamon. This is in view of developing a safe, readily soluble Cinnamon extract for human consumption that could effectively control hyperglycemia in diabetic individuals. Hence in the current study, the total phenolic content, proanthocyandins and essential oils were determined as compounds largely responsible for the biologicals activities explored. The results showed that Sri Wijaya Cinnamon water extract prepared using high pressure and the decoction method had the highest anti-diabetic potential. The extracts have to be further purified and developed with in depth studies using cell cultures and experimental animals for the treatment of diabetes mellitus.

## ACKNOWLEDGEMENTS

The authors wish to thank Mr. A.T. Kannangara of the Department of Chemistry, University of Kelaniya, Sri Lanka for his assistance in GC/MS analysis.

### Funding

This work was supported by the National Science Foundation of Sri Lanka (SP/CIN/2016/03). The funders had no role in study design, data collection and analysis, decision to publish, or preparation of the manuscript.

### Grant Disclosures

The following grant information was disclosed by the authors:
National Science Foundation of Sri Lanka: SP/CIN/2016/03.

### Competing Interests

The authors declare there are no competing interests.

### Author Contributions

- W.A. Niroshani M. Wariyapperuma conceived and designed the experiments, performed the experiments, analyzed the data, prepared figures and/or tables, authored or reviewed drafts of the paper, and approved the final draft.
- Sagarika Kannangara conceived and designed the experiments, performed the experiments, authored or reviewed drafts of the paper, and approved the final draft.
- Yasanandana S. Wijayasinghe and Sri Subramanium conceived and designed the experiments, authored or reviewed drafts of the paper, and approved the final draft.
- Bimali Jayawardena conceived and designed the experiments, performed the experiments, analyzed the data, authored or reviewed drafts of the paper, and approved the final draft.

### Data Availability

The raw measurements are available in the Supplemental Files.

The voucher specimen of "Sri Wijaya" and "Sri Gemunu" accessions were deposited at the publicly available herbarium, Department of Plant and Molecular Biology, the University of Kelaniya, Sri Lanka under the family Lauraceae (Deposition numbers are "CIN-SW-001" and "CIN-SG-002" respectively for "Sri Wijaya" and "Sri Gemunu" accessions).

### Supplemental Information

Supplemental information for this article can be found online at http://dx.doi.org/10.7717/peerj.10070#supplemental-information.

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
