# Peer review of "In vitro anti-diabetic effects and phytochemical profiling of novel varieties of Cinnamomum zeylanicum (L.) extracts"

_PeerJ, doi:10.7717/peerj.10070_

## Round 0.1 · original submission · Major Revisions

Please consider carefully all questions raised up by the reviewers, particularly those regarding the design of the study and the connection between the conclusions and the hypothesis of the study.

·

Basic reporting

Dear editor-in-chief,
Thank you for sending the paper for peer review.
Paper referred to the diabetes and phytochemical profiling of novel varieties of Cinnamomum zeylanicum (L.) extracts. They conducted chemical analysis. They found the potential antidiabetic properties of the Sri Wijaya Cinnamon accessions, which exhibited significant α-amylase and α-glucosidase inhibitory activities.

Paper, is suitable for publication. I checked all pages. However, a plagiarism check is also necessary. They talked on the anti-diabetic impact of cinnamon, however, this medicinal plant had anti-hypertensive effect too. There, I would like the authors to read and cite and use in the text, the following article;
Mahmoodnia L, Aghadavod E, Rafieian-Kopaei M. Ameliorative impact of cinnamon against high blood pressure; an updated review. J Renal Inj Prev. 2017;6(3):171-176.

Experimental design

N/A

Validity of the findings

Good

·

Basic reporting

See general comments below

Experimental design

See general comments below

Validity of the findings

See general comments below

Additional comments

Review of manuscript title: In vitro anti-diabetic effects and phytochemical profiling of novel varieties of Cinnamomum zeylanicum (L.) extracts by Wariyapperuma Appuhamillage Niroshani M Wariyapperuma et al.

General comment
The study attempts to identify the potential antidiabetic properties of the Sri Wijaya Cinnamon varieties with significant α-amylase and α-glucosidase inhibitory activities. Three different varieties of cinnamon submitted to distint extraction methods were prepared and tested for their enzyme inhibitory activity. The results showed that extracts prepared using pressured water and as a decoctions were more potent in inhibiting α-amylase and α-glucosidase activity, and had the potential to extract higher content of proanthocyanidin than other methods of extraction. The hypothesis is quite attractive, experimental design is sound and methods seem adequate, in particular the exhaustive chemical characterization of the extract; results are clearly presented and discussed and look promissing for a future utilization in humans. Search for plant foods and their specific components which might be relevant in a nutritionnal prevention and coadjuvant treatment of type 2 diabetes is a high demanded topic in current nutritional biochemistry and studies dealing with this subject should be welcome. Some specific comments are detailed below:

Specific comments
1) Lines 110-130, line 156 and lines 249-250; it should be mL, with capital L, as in the rest of the manuscript.

2) Figure 1 is missing statistical symbols; although in line 182 is reported that there were significant differences among extract yields and extraction methods, there is no indication of such statistical differences in the figure or figure legend.

3) Line 252; it should say Metformin.

4) Line 256; proanthocyanidins are considered as a subgroup or subfamily of polyphenols, perhaps it would be more correct to state that polyphenols, in particular proanthocyanidin polimers…

·

Basic reporting

The article is clearly written and structured in the corresponding sections.
Only minor grammar mistakes have been marked in the text.
Please, keep consistency when alfa-amylase and alfa-glucosidase are written throughout the text: use always the same criteria: capital letters for the first "a" and "g", or not.
Literature references are correct and updated.
Figures and tables are correct and justified.
Results are considered relevant but they don't support the hypotheses or the conclusions of the work: authors stated that benzoic acid, cinnamyl alcohol, banzyl alcohol and the derivative from dimethoxyphenol, which are identified as the major compounds in the extracts, are believed to be responsible of the inhibitory activity. Nonetheless, these compounds are part of the essential oil composition and authors do no discuss any research work proving the cited activity in any research model. What about the polyphenols composition? Only the total amount has been calculated, not the kind of specific polyphenols: flavonoids, phenolic acids… which are widely known as strong active compounds

Experimental design

The research is well designed and fulfills the aims and scope of the journal.
The research question is well defined and the results are relevant, but do not support the conclusions held by the authors: phytochemical profile of the essential oil by GC is really interesting but does not prove any relationship with the in vitro activity.
What about the polyphenols?

Methods are correctly described

Validity of the findings

The findings are correct and relevant for the field of research, but they are limited to support the conclusions of the authors.
The performed in vitro studies are not enough to support the treatment of diabetes mellitus, as stated on lines 320-321. Previous in depth studies in cell cultures and experimental animals should be conducted to confirm the observed tendency and to identify the compounds which are responsible.

Reviewer 4 ·

Basic reporting

Though Cinnamomum has been extensively researched, the topic was very interesting and curiosity pulling to know in depth. Here some of my points of view:
- The English writing is clear and good enough to understand the content.
- The obtaining plants accession needs more botanical or genetical background justification. You must be clearly classified what do you mean by Cinnamon accession, is it variety or subculture category. The business source information https://www.lankabusinessonline.com/sri-lanka-develops-75 new-Cinnamon-varieties which could not be opened yet, should not be your main information according to your material source.
- so did with the reason why have to take so many varieties on the extraction.

Experimental design

- The research question is not well defined. There was steam distillation followed by oleoresin n-hexane extraction which not equally compare to many kinds of water extraction method.
- critical important variables were not be controlled (the material weight for extraction, the temperature and the quantity of solvent used)… SD (40 g) while others 10 g – line 93-97
- no standard/ guidance on the extraction method (SE line 98, DWE, IWE line 102-103)
- The extractive method produced non-volatile compounds most. The GC-MS is not suitable for analysis components, except for SD.

Validity of the findings

- The SE and PWE of SW & SG yields seem wide SD, so did with SE on α-glucosidase and seem need be refined further.

Additional comments

To improve your finding, please describe the method precisely, including the limitation of the research
here some references related:
-Marongiu B, Piras A, Porcedd S, Tuveri E, Sanjust E, Meli M, Sollai F, Zucca P, and Rescigno A. Supercritical CO2 extract of Cinnamomum zeylanicum: chemical characterization and antityrosinase activity. J. Agric. Food Chem. 2007 2007, 55, 10022–10027
-Ervina M, Lie HS, Diva J, Caroline, Tewfik S, Tewfik I. Optimization of water extract of Cinnamomum burmannii bark to ascertain its in vitro antidiabetic and antioxidant activities. Biocatalysis and Agricultural Biotechnology 19 (2019) 101152
-Kasim NN, Ismail SNAS, Masdar ND, Ab Hamid F, Nawawi WI. Extraction and Potential of Cinnamon Essential Oil towards Repellency and Insecticidal Activity. International Journal of Scientific and Research Publications, 2014 (4.7)
Wardatun S, Rustiani E, Alfiani N, Rissani D. Study Effect Type of Extraction Method And Type of Solvent To Cinnamaldehyde and Trans-Cinnamic Acid Dry Extract Cinnamon (Cinnamomum burmanii [Nees & T, Nees]Blume). J Young Pharm, 2017;9(1)Suppl: s49-s51

---

## Round 0.2 · Minor Revisions

I think the authors have properlly addressed most comments raised up by the reviewers. Nevertheless, in order to make it acceptable for publication, some rearrangments should be performed in the Concusions. Thus, the sentence on essential oils should be placed before authors start to provide their own results. And they should make clear that, in their study, essential oils profile was also assessed.

---

## Round 0.3 · Minor Revisions

I think the revised Conclusions are not completely clear. Therefore, I would suggest this alternative writing:

"Whilst there are numerous studies on the biological activity of essential oils, aspects such as safety, astringency, cost and the solubility in water stifle their applications as nutraceuticals. This study was mainly focused on investigating the in vitro hypoglycemic effects of aqueous extracts of Cinnamon. This is in view of developing a safe, readily soluble Cinnamon extract for human consumption that could effectively control hyperglycemia in diabetic individuals. Hence in the current study, the total phenolic content, proanthocyandins and essential oils were determined as compounds largely responsible for the biologicals activities explored. The results showed that Sri Wijaya Cinnamon water extract prepared using high pressure and the decoction method had the highest anti-diabetic potential. The extracts have to be further purified and developed with in depth studies using cell cultures and experimental animals for the treatment of diabetes mellitus"

---

## Round 0.4 · accepted · Accept

After modifying the Conclusions, I think the manuscript is now acceptable for publication.